# MRI Radiomics-Based Machine Learning Models for Ki67 Expression and Gleason Grade Group Prediction in Prostate Cancer

**DOI:** 10.3390/cancers15184536

**Published:** 2023-09-13

**Authors:** Xiaofeng Qiao, Xiling Gu, Yunfan Liu, Xin Shu, Guangyong Ai, Shuang Qian, Li Liu, Xiaojing He, Jingjing Zhang

**Affiliations:** 1Department of Radiology, The Second Affiliated Hospital of Chongqing Medical University, Chongqing 400010, China; qiaoxiaofeng@hospital.cqmu.edu.cn (X.Q.); guxiling@cqmu.edu.cn (X.G.); szpm515@163.com (Y.L.); bryant0824sx@163.com (X.S.); 303701@hospital.cqmu.edu.cn (G.A.); 2Big Data and Software Engineering College, Chongqing University, Chongqing 400000, China; 202024131068@cqu.edu.cn (S.Q.); dcsliuli@cqu.edu.cn (L.L.); 3Departments of Diagnostic Radiology, National University of Singapore, Singapore 119074, Singapore; 4Clinical Imaging Research Centre, Centre for Translational Medicine, National University of Singapore, Singapore 117599, Singapore

**Keywords:** prostate cancer, MRI, Gleason grade group, Ki67, machine learning

## Abstract

**Simple Summary:**

Given the variable aggressiveness of PCa, patients with indolent PCa do not require intervention, but rather require active surveillance and close lifelong follow-up, while those with invasive PCa require surgery, various types of radiation therapy, androgen-deprivation therapy (ADT), or multimodal treatment. Hence, it is critical to accurately distinguish indolent from invasive PCa for prognosis evaluation and treatment decision-making. The aim of the present study was to investigate the value of MR radiomics feature-based machine learning (ML) models in predicting the Ki67 index and Gleason grade group (GGG) of PCa. Biparametric magnetic resonance imaging (bpMRI) radiomics-based ML models to predict immuno-histochemically-determined Ki67 expression and the GGG demonstrated the ability to identify aggressive PCa. A preliminary exploration was performed in the conjoint analysis, laying the theoretical foundation for models predicting two or more variables; such models are expected to provide more comprehensive pathological information and provide valuable guidance for clinical decision-making in a noninvasive, synchronous, and objective manner.

**Abstract:**

Purpose: The Ki67 index and the Gleason grade group (GGG) are vital prognostic indicators of prostate cancer (PCa). This study investigated the value of biparametric magnetic resonance imaging (bpMRI) radiomics feature-based machine learning (ML) models in predicting the Ki67 index and GGG of PCa. Methods: A total of 122 patients with pathologically proven PCa who had undergone preoperative MRI were retrospectively included. Radiomics features were extracted from T2-weighted imaging (T2WI), diffusion-weighted imaging (DWI), and apparent diffusion coefficient (ADC) maps. Then, recursive feature elimination (RFE) was applied to remove redundant features. ML models for predicting Ki67 expression and GGG were constructed based on bpMRI and different algorithms, including logistic regression (LR), support vector machine (SVM), random forest (RF), and K-nearest neighbor (KNN). The performances of different models were evaluated with receiver operating characteristic (ROC) analysis. In addition, a joint analysis of Ki67 expression and GGG was performed by assessing their Spearman correlation and calculating the diagnostic accuracy for both indices. Results: The ML model based on LR and ADC + T2 (LR_ADC + T2, AUC = 0.8882) performed best in predicting Ki67 expression, and ADC_wavelet-LHH_firstorder_Maximum had the highest feature weighting. The SVM_DWI + T2 (AUC = 0.9248) performed best in predicting GGG, and DWI_wavelet HLL_glcm_SumAverage had the highest feature weighting. The Ki67 and GGG exhibited a weak positive correlation (*r* = 0.382, *p* < 0.001), and LR_ADC + DWI had the highest diagnostic accuracy in predicting both (0.6230). Conclusion: The proposed ML models are suitable for predicting both Ki67 expression and GGG in PCa. This algorithm could be used to identify indolent or invasive PCa with a noninvasive, repeatable, and accurate diagnostic method.

## 1. Introduction

Prostate cancer (PCa) accounts for an estimated 29% of all new incident cases and has become the second leading cause of cancer-related deaths (11%), according to the American Cancer Society (ACS) [1]. Given the variable aggressiveness of PCa, patients with indolent PCa do not require intervention, but rather require active surveillance (AS) and close lifelong follow-up, while those with invasive PCa require surgery, various types of radiation therapy, androgen-deprivation therapy (ADT), or multimodal treatment [2,3,4]. Hence, it is critical to accurately distinguish indolent from invasive PCa for prognosis evaluation and treatment decision-making.

The Gleason score (GS), as a widely recognized indicator of PCa aggressiveness, reflects the differentiation degree and heterogeneity of prostate tumor cells [5,6,7]. Tumors with lower GSs are predominantly associated with indolent PCa, whereas tumors with higher GSs exhibit greater aggressiveness due to dysplastic, fused, or cribriform glands. The new Gleason grade group (GGG) scale reflects PCa biology and predicts tumor progression more accurately than the Gleason score system, according to the International Society of Urological Pathology (ISUP). Studies have demonstrated that the hazard ratios of GGG 2, 3, 4, and 5 relative to GGG 1 are 2.2, 7.3, 12.3, and 23.9, respectively, and the corresponding 5-year biochemical risk-free survival rates are 96%, 88%, 63%, 48%, and 26%. The GGG is an important indicator for prognostic assessment, treatment regimen formulation, and survival prediction in PCa [7].

The nuclear protein Ki67 is a nonspecific marker of cell proliferation expressed only in the cell proliferation cycle, which contributes to evaluating the tumor growth fraction [8]. The assessment of Ki67 has made great contributions to the evaluation of tumor proliferation and invasion, especially in breast cancer, since it was first applied to lymphoma in 1984. Higher Ki67 expression is strongly associated with a higher GS, more advanced cancer, seminal vesicle invasion, extracapsular extension, poorer survival, and a higher risk of fatal cancer [9,10,11,12,13,14,15,16,17]; Matthew K. Tollefson et al. found that, in PCa, the risk of disease-specific death increased by nearly 12% with each 1% increase in Ki67 expression after adjusting for perineural invasion and GS [18].

Although Ki67 expression and the GGG are important indicators for assessing tumor invasiveness, there are limitations to their independent application. For instance, PCa is usually a multifocal tumor, and the GS is determined by the highest score among the tumors, which may lead to an overestimation of the overall GGG. Ki67 evaluation alone is not recommended for outcome prediction due to the high heterogeneity and variability of PCa, although it has great value in low-risk/indolent PCa [13]. Ki67 is, however, considered a good supplement to the GS [14,15,16]. The combination of Ki67 and GSs may be the best indicator for long-term PCa outcome assessment [18] and may offer more comprehensive preoperative information. Nevertheless, both the GGG and Ki67 expression are obtained through histopathology, which requires an invasive procedure that may result in complications such as postsurgical hemorrhage, pain, infection, and prolonged hospital stay. Therefore, noninvasive methods for Ki67 and GGG assessment need to be explored. Biomarker analysis and magnetic resonance imaging (MRI) were recommended as alternatives to conventional biopsy at the 23rd Annual Conference of the National Comprehensive Cancer Network (NCCN) [19].

Considering the demonstrated advantages of MRI, it is widely used for tumor characterization, staging, treatment planning, targeted therapy, treatment response assessment, and surveillance. However, interobserver differences are a nonnegligible limitation of the technique [20]. With the development of artificial intelligence, radiomics combined with machine learning (ML) has become a novel approach in medical image analysis, by which a large amount of high-quality and quantitative data are obtained noninvasively and objectively. Therefore, radiomics-based ML has emerged as a field of high research interest in recent years, and great achievements have been made in numerous areas, especially in the assessment of malignant tumors [21,22,23,24,25,26,27]. There have been relatively few studies on predicting Ki67 and GGG. Duc Fehr et al. presented an ML-based automatic classification of GS that combines apparent diffusion coefficient (ADC)- and T2-weighted imaging (T2WI)-based texture features [28]. Fan et al. constructed innovative ML models for predicting Ki67 based on T2WI, diffusion-weighted imagine (DWI) and dynamic contrast-enhanced (DCE) MRI, which contributed to the risk stratification evaluation of PCa [29]. However, most studies to date have focused on the use of single clinical or pathological variables. Considering the great clinical value of the GGG combined with Ki67, the ultimate goal of this study is to shift the prognostic assessment and treatment decision-making paradigm in PCa from an invasive, subjective process to a noninvasive, objective process by predicting Ki67 expression and the GGG through ML-based image analysis.

## 2. Materials and Methods

### 2.1. Patients

The institutional review board of our hospital approved this retrospective study and waived the requirement for informed consent (decision number (2019) 289). Patients who underwent prostate MRI scanning between August 2016 and May 2021 were screened for inclusion according to the following criteria: (1) PCa confirmed by pathology, (2) sufficient tissue samples for immunohistochemical Ki67 and GGG analyses, and (3) systematic biopsy performed within 3 months after the MRI examination. The exclusion criteria were as follows: (1) operation or endocrine treatment before MRI examination (*n* = 16) and (2) poor MR image quality (*n* = 2). The details are summarized in Figure 1.

### 2.2. MRI Protocol

The biparametric magnetic resonance (bpMR) images were acquired using a 3.0 T MRI scanner (MAGNETOM Prisma; SIEMENS A Tim Dot System) and an 8-channel phased-array software coil. The scan covered the prostate gland, the seminal vesicles, and as many adjoining structures as possible. The parameters for fat suppression (FS) axial T2WI were as follows: repetition time (TR)—3090 ms; echo time (TE)—77 ms; slice thickness—3 mm; number of excitations (NEX)—2; field of view (FOV)—20 × 20 cm; and acquisition matrix—320 × 240. Axial DWI was performed in the same orientation and location as in axial T2WI using axial echo-planar imaging (EPI) sequences as follows: TR—3800 ms; TE—84 ms; slice thickness—3 mm; NEX—2; FOV—20 × 20 cm; acquisition matrix—118 × 118; and b value—1400 s/mm^2^. ADC maps were automatically generated from intravoxel incoherent motion (IVIM) imaging on a designated workstation (Version syngo MR E11; Siemens software packages (syngo.via VB20A_HF06); NUMARIS/4), in which the low b-value was 0 s/mm^2^ and the intermediate b-value was 1000 s/mm^2^.

### 2.3. Pathology

All patients underwent transrectal ultrasound (TRUS)-guided 12-core systematic biopsy within 3 months after the MRI examinations. The Ki67 expression of all prostatic parenchymal tissue samples was re-evaluated by a pathologist with more than 15 years of experience. To determine the immunohistochemical evaluation standard for Ki67, the percentage of stained cells in three hotspots (the areas with the most intense proliferation) was calculated, and the average value was taken. When there were multiple tumors, the maximum value was taken [16,17]. The data were classified into 2 subgroups in task 1 and task 2 according to the pathological results: (1) Task 1_Ki67—the patients were divided into a high-expression group (>10%) and a low-expression group (≤10%), using the median value as the cutoff [12,29]; (2) Task 2_GGG—the patients were classified into low-grade (GGG 1-2) and high-grade (GGG 3-5) groups according to strict evaluation using the Gleason Grade Conference standards of the 2019 ISUP [7].

### 2.4. Radiomics Feature Extraction

All original T2W, DW, and ADC images, stored in DICOM format, were imported into Artificial Intelligence Kit software (A.K. software; GE Healthcare) for delineating the region of interest (ROI) of tumor areas and extracting features. Two doctors (a radiologist with 5 years of experience in abdominal imaging and an attending doctor with 10 years of experience in urinary imaging) read the original images together and then drafted the lesion ROI segmentation plan by discussion until a consensus was reached. If any disagreements could not be resolved, the final judgment was conducted by a third reviewer with extensive seniority, possessing over 15 years of experience in urinary system imaging. The ROI delineation encompassed the entire tumor while excluding the invasion of the urethra and adjacent structures. For multifocal tumors, only the largest lesion was segmented. An example of ROI segmentation and the corresponding pathological images (GGG and Ki67) are shown in Figure 2. Following the application of gray-level discretization with a bin size of 5, a total of 107 radiomics features were extracted, including first-order features (*n* = 18), shape features (*n* = 14), and texture features (*n* = 75); the latter, used to describe the internal and surface textures of the lesions, consisted of the gray-level cooccurrence matrix (GLCM) (*n* = 24), the gray-level size zone matrix (GLSZM) (*n* = 16), the gray-level dependence matrix (GLDM) (*n* = 5), the gray-level run length matrix (GLRLM) (*n* = 16), and the neighborhood gray tone difference matrix (NGTDM) (*n* = 14) features. Subsequently, by employing wavelet transformation (level = 1), 851 radiomic features were extracted from each of the T2WI, DWI, and ADC images. In total, 2553 radiomic features were ultimately obtained.

### 2.5. Preprocessing of Radiomic Features

The data were normalized, scaled, and dimensionally reduced for subsequent comparison. Recursive feature elimination (RFE) was conducted to eliminate redundant and irrelevant features by fitting a given ML algorithm used in the core model, computing and ranking the importance of each feature, and discarding the least important features. This process was repeated until the optimal combination of radiomic features was obtained. Following feature elimination, each model was constructed using a selected set of 20 features. The synthetic minority oversampling technique (SMOTE) was used to resolve data imbalance problems, and 5-fold cross-validation was applied to validate the performance of the ML models. The dataset was randomly divided into a training and testing set with an 8:2 ratio. The code for models training and testing was implemented on Python 3.6, and the common libraries included numpy 1.19.2, cv 2 4.5.1, SimpleITK 2.0.2, pandas 1.1.5, and scikit-learn 0.24.2. This process was repeated five times, and the average of the five iterations was taken as the final experimental result of this study.

### 2.6. Construction of ML Models

Twenty-eight ML models were established to predict Ki67 expression and GGG based on four ML algorithms and seven different sequences or sequence combinations. The classification algorithms included logistic regression (LR), support vector machine (SVM), random forest (RF), and K-nearest neighbor (KNN). The sequences included T2WI, DWI, ADC, and 4 combinations thereof (T2+DWI, T2+ADC, ADC+DWI, ADC+DWI+T2). Subsequently, the feature compositions of the best models for the two tasks were analyzed. A technical flowchart of this study is presented in Figure 3.

### 2.7. Statistical Analysis

All statistical analyses were performed with SPSS software (version 25; IBM Corporation, Armonk, NY, USA) and Python codes. Comparisons among groups were performed by the independent-sample *t*-test. Normally distributed variables are shown as the mean ± standard deviation (SD). Non-normally distributed variables are expressed as the median with the interquartile range in parentheses (25th and 75th percentiles). *p* < 0.05 was considered statistically significant. The performance of the ML models was evaluated by using receiver operating characteristic (ROC) curve analysis, including calculation of the area under the curve (AUC), sensitivity, and specificity. The Spearman correlation test was applied to analyze the correlation between Ki67 expression and GGGs. Finally, a joint analysis of the two tasks was performed by calculating the diagnostic accuracy.

## 3. Results

### 3.1. Baseline Characteristics

A total of 122 patients were enrolled in this study, including 31 with tumors located in the peripheral zone (PZ), 27 with tumors located in the transition zone (TZ), and 64 with cross-zone tumors, occupying both the PZ and TZ. For task 1, the Ki67 ≤ 10% group included 67 patients (average age 72.7 ± 8.6 years), and the Ki67 > 10% group included 55 patients (average age 71.4 ± 8.7 years). For task 2, the GGG 1-2 group had 37 patients (average age 72.4 ± 8.8 years), and the GGG 3-5 group had 85 patients (average age 72.0 ± 8.6 years). There was no statistically significant age difference between the subgroups in either task 1 (*t* = 0.801, *p* = 0.425) or task 2 (*t* = 0.213, *p* = 0.832). The median (25th–75th percentile) prostate-specific antigen (PSA) and free PSA (fPSA) levels were 47.3 (16.4–136.5) ng/mL and 4.00 (1.37–13.35) ng/mL, respectively.

### 3.2. Task 1: ML Models for Predicting Ki67 Expression

The comparisons of the 28 ML models are displayed in Table 1 and Figure 4. LR had the best performance, especially for the combination sequences. LR_T2+ADC (AUC = 0.8882, sensitivity = 0.7636, specificity = 0.8657) achieved the optimal performance among the 28 ML models.

The radiomics features of LR_T2+ADC are illustrated in Figure 5. Of the 20 features, 11 were derived from ADC imaging, and the other 9 were from T2WI, including 7 first-order features, 2 shape features, and 11 texture features. Among them, ADC_wavelet-LHH_first-order_Maximum had the highest weight in predicting Ki67 expression.

### 3.3. Task 2: ML Models for Predicting GGG

The comparison results for the four ML algorithms and seven sequences are displayed in Table 2 and Figure 6. SVM and LR performed better than KNN and RF. Among the 28 ML models, the optimal model was SVM_T2+DWI (AUC = 0.9248, sensitivity = 0.8588, specificity = 0.7838).

The radiomics feature analysis of SVM_T2 + DWI is presented in Figure 7. The 20 radiomics features consisted of 4 first-order features and 16 texture features, with 18 from DWI and 2 from T2WI. The radiomics feature with the highest weight was DWI_wavelet-HLL_glcm_SumAverage.

### 3.4. Conjoint Analysis of the Ki67 and GGG Tasks

The Ki67 and GGG exhibited a weak positive correlated (*r* = 0.382, *p* < 0.001). Then, the diagnostic accuracies of the combined task (task 1 and task 2) were calculated. LR_ADC + DWI performed best, with an accuracy of 0.6230, in making correct predictions in both task 1 and task 2, as illustrated in Table 3.

A detailed feature comparison for LR_ADC+DWI in the combined task is shown in Figure 8. The feature comparison consisted of four parts, which yielded the following findings. First, the features were extracted from both ADC (*n* = 6) and DWI (*n* = 14) in two tasks with the same extraction ratio. Second, most of the features were subjected to a wavelet transform (Ki67 = 16/20, GGG = 19/20). Third, the proportions of first-order, shape, and texture features are similar (Ki67 = 3/1/16, GGG = 4/1/15). Fourth, seven features were shared between the two tasks.

## 4. Discussion

In this study, we constructed bpMRI radiomics-based ML models to predict Ki67 expression and GGGs in PCa, and we performed a conjoint analysis of the two corresponding tasks, establishing a foundation for the prediction of multiple pathological indicators. The results demonstrated that the best model for predicting Ki67 expression (task 1) was LR_T2+ADC, the best model for predicting GGGs (task 2) was SVM_T2+DWI, and the highest-performing model in the conjoint analysis was LR_ADC+DWI. These bpMRI radiomics ML models could be applied in clinical practice to obtain pathology information noninvasively and objectively for PCa assessment and treatment-related decision-making.

Four ML algorithms were used in this study to predict Ki67 and GGGs. Overall, the LR- and SVM-based models were superior to the KNN- and RF-based models. LR (AUC = 0.8882) was the best-performing classification algorithm in task 1. However, Xuhui Fan et al. found that RF was the best-performing algorithm in their study (AUC = 0.87) [29]. We speculate that the reason for this discrepancy is that LR, as a supervised classifier, is adept at binary classification tasks, and the training result is mainly influenced by feature weights. The use of adequate high-weight radiomic features in our study might have improved the performance of the LR model. This speculation was confirmed by Lili Zhou et al. [30], who indicated that LR yielded superior results when the key information in the sample was sufficient for predicting Ki67 in medulloblastoma. SVM (AUC = 0.9248) had the best predictive performance in task 2, consistent with the results from D.F. and H.V. et al. [28] SVM considers the interactions among all features and removes features whose effects might offset each other during each iteration in the training until the best performance is achieved.

The images of bpMRI (T2, DWI, ADC) were included individually and in different combinations in this study to comprehensively evaluate the predictive ability of the models. T2+ADC (AUC = 0.8882) was the highest-performing model in task 1, followed by ADC+DWI+T2 (AUC = 0.8673). In a study by Xuhui Fan et al., the T2+DWI+DCE-based model (AUC = 0.87) significantly outperformed the T2 + DWI model (AUC = 0.8285) [29]. We speculate that DCE exerts a large effect associated with neovascularization in PCa. However, our T2+ADC-based model achieved comparable or even better performance without the use of contrast agents. One potential explanation for this outcome could be the fourth dimension (time) present in DCE, which poses challenges in aligning and matching two-dimensional anatomical images like T2WI and DWI. Additionally, extracting information on contrast media arrival and distribution, serving as a surrogate marker for microvascular density, requires the use of semi-quantitative or quantitative pharmacokinetic models, thereby introducing further complexity in post-processing [31]. BpMRI offers the advantage of avoiding potential harmful effects associated with gadolinium contrast agents while also enhancing confidence in the additional value of bpMRI in radiomics research [32]. In contrast to the results for the Ki67 task, the T2+DWI (AUC = 0.9248) was the highest-performing model in the GGG task. Previous studies have placed greater emphasis on T2 and ADC sequences [28,33,34,35,36]. Ahmad Chaddad et al. constructed a predictive model for predicting GS based on T2+ADC, with an AUC value of 0.8235 [34]. We surmise that finer grouping may lead to slightly decreased performance. On the other hand, most of the features were from DWI (*n* = 18) rather than T2 (*n* = 2) in this study. One of the reasons might be that some heavily weighted radiomic features from DWI better represented the characteristics of cell atypia and increased tumor cell density with increasing GGG. Frustratingly, ADC+DWI+T2 had the worst performance in predicting GGG, even lower than that of any of the individual sequences. A potential explanation is that the higher-weight features might cancel each other out, but this possibility remains to be studied thoroughly.

We further analyzed the radiomic features and found that wavelet features had a particularly high proportion among the best models for all tasks (≥80%), similar to the research of Xu Huifan et al. [29]. Another study on rectal cancer reported that wavelet features accounted for the highest proportion of features in models predicting Ki67 (75%), HER-2 (70%), and lymph node metastasis (80%) [37]. It appears to us that wavelet features have strong predictive ability, effectively representing tumor heterogeneity. In the feature-type analyses, texture features were the most common, comprising 55.5% of all features for the Ki67 task, 80% for the GGG task, and 80% for the Ki67+GGG task. First-order and shape features represent the distribution of voxel intensities and 3D shapes, which are more or less reliant on manual segmentation, an inevitable source of bias. Studies have demonstrated that texture features can be used to accurately classify GGGs [34] and can be stably extracted from the original image.

The best ML algorithms in the conjoint analysis were LR and SVM, but the highest accuracy was only 0.6230. We speculate that the reason may be as follows. First, Ki67 and GGG are excellent predictors of the invasiveness of PCa, but they are essentially different, with Ki67 representing the proliferative capacity of tumor cells and GGG representing cell atypia and differentiation capacity. The five texture features that overlapped between the two tasks (SmallAreaLowGrayLevelEmphasis, Maximum, Minimum, ZoneEntropy, and Skewness) are the most frequently mentioned features in the literature [28,29,33,34]. Second, the complexity of the calculation increased when the tasks were superimposed. Particularly in patients with high Ki67/low GGG and low Ki67/high GGG, even small changes in proportions may affect the calculations of the models. Much more research and analysis is necessary to achieve the prediction of two or more variables.

This study had some limitations that should be noted. First, ROI segmentation could be optimized in follow-up studies using semiautomatic or automatic methods instead of manual segmentation. Second, the pathological specimens in this study were derived from systematic biopsy, which might lead to underestimation, sampling errors, and bias. This shortcoming may be improved by performing radical prostatectomy and targeted biopsy. Third, while Ki67 and GGG are of significant importance in preoperative and prognostic assessments, their inherent limitations and inability to encompass all clinical information suggest that utilizing 5-year overall survival (OS) or disease-free survival (DFS) as prognostic indicators might be a more optimal choice. Furthermore, our study received a radiomics quality score (RQS) of 8 according to the assessment [38], which is comparable to the reported score for prostate imaging (7.93) [39] and slightly higher than the general one (7.56) [40]. However, the main factor affecting the RQS of this study is the lack of external validation. By introducing external validation, the credibility of this single-center retrospective observational study can be significantly enhanced. Each of these points is worthy of further study.

## 5. Conclusions

In this study, nonenhanced MRI radiomics-based ML models that were used to predict immunohistochemically determined Ki67 expression and the GGG demonstrated the ability to identify aggressive PCa. A preliminary exploration was performed in the conjoint analysis, laying the theoretical foundation for models predicting two or more variables; such models are expected to provide more comprehensive pathological information and provide valuable guidance for clinical decision-making in a noninvasive, synchronous, and objective manner.

## Figures and Tables

**Figure 1 cancers-15-04536-f001:**
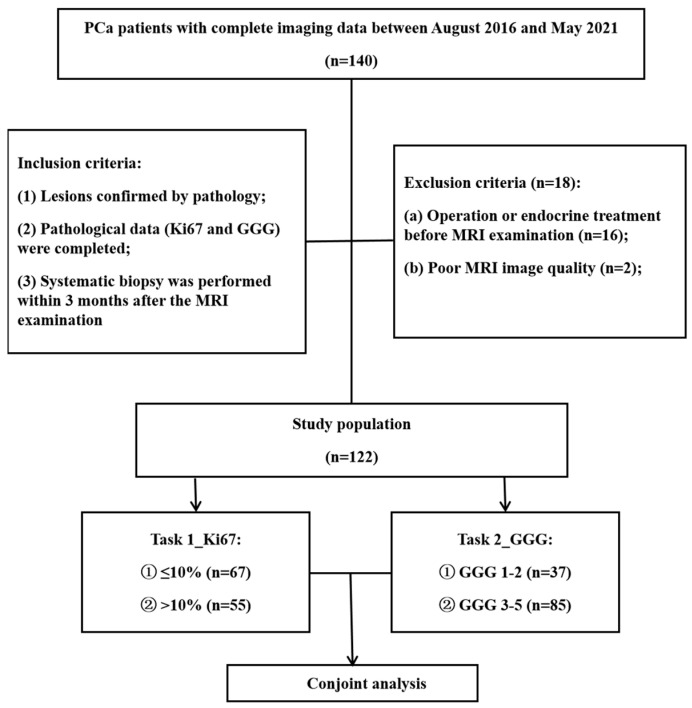
Flowchart of patient selection. PCa = prostate cancer, GGG = Gleason grade group.

**Figure 2 cancers-15-04536-f002:**
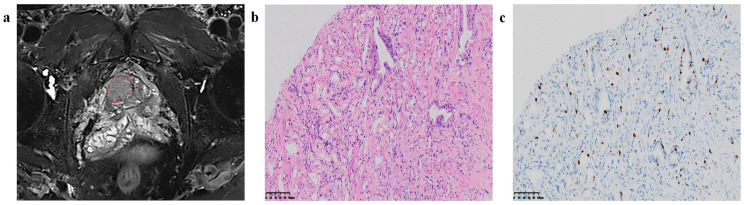
Region of interest (ROI) segmentation and corresponding pathological pictures. (**a**) ROI segmentation, (**b**) Loupe image of hematoxylin–eosin stain shows the GS = 4 + 3 (×20), (**c**) Loupe image of immunohistochemical stain shows the percentage.

**Figure 3 cancers-15-04536-f003:**
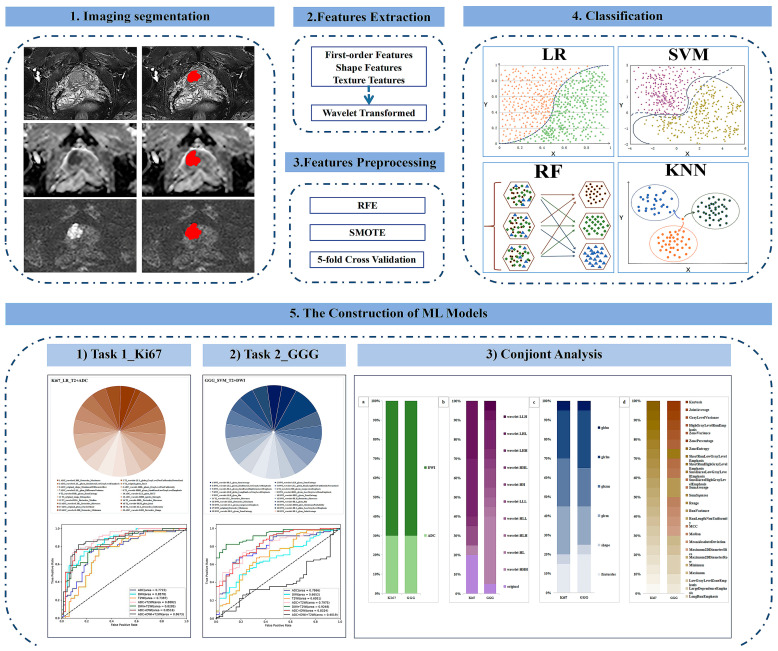
Technical flowchart. RFE = recursive feature elimination, SMOTE = synthetic minority oversampling technique, LR = logistic regression, SVM = support vector machine, RF = random forest, KNN = K-nearest neighbor, ML = machine learning, GGG = Gleason grade group.

**Figure 4 cancers-15-04536-f004:**
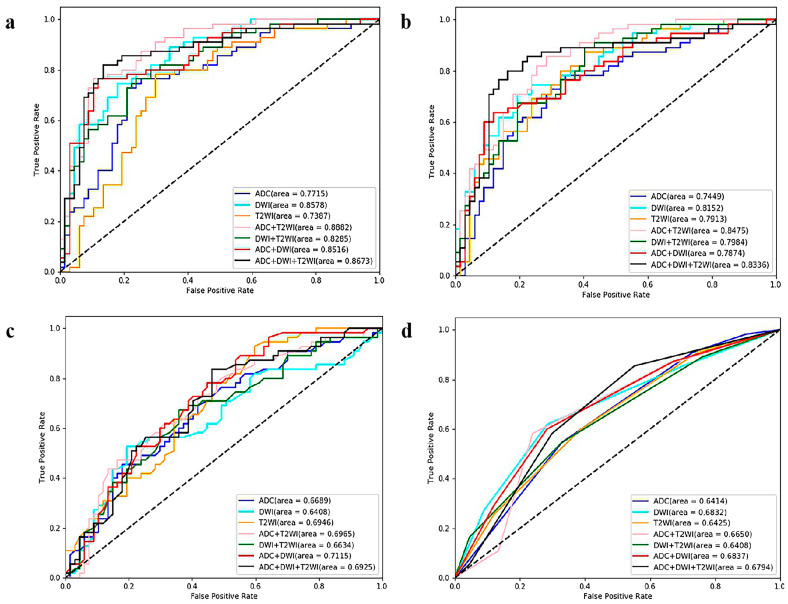
ROC curves of the twenty-eight ML models (based on four algorithms and seven sequences) for predicting Ki67 expression *. (**a**) Logistic regression-, (**b**) support vector machine-, (**c**) random forest-, and (**d**) K-nearest neighbor-based model ROC curves. * ROC = receiver operating characteristic, ML = machine learning, T2WI = T2-weighted imaging, DWI = diffusion-weighted imaging, ADC = apparent diffusion coefficient.

**Figure 5 cancers-15-04536-f005:**
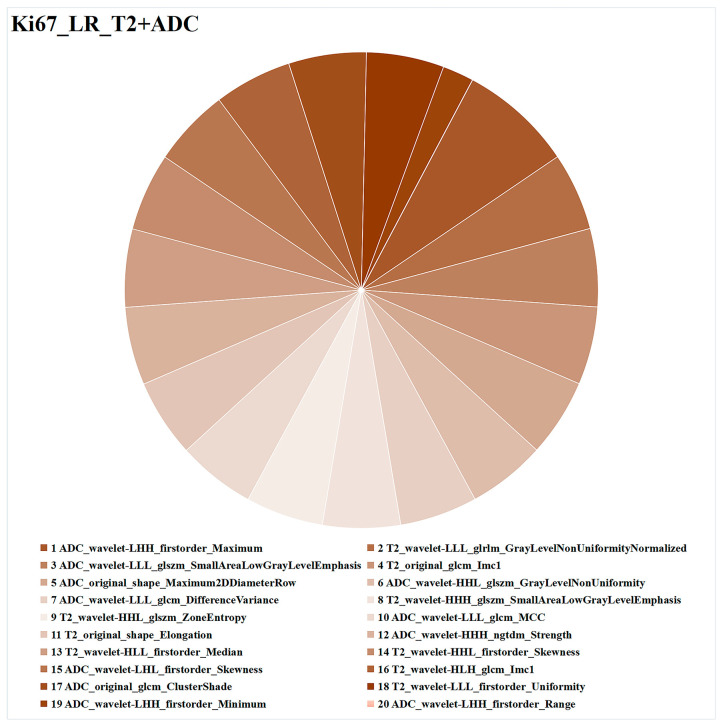
Optimal combined ML model (LR_T2+ADC) for task 1 (Ki67 expression prediction)**.** Different colors represent different radiomics features, and the area represents the weighted contribution of each radiomics feature in the ML model. The feature with the highest weight for Ki67 expression prediction was ADC_wavelet-LHH_first-order_Maximum. ML = machine learning, LR = logistic regression, T2 = T2-weighted imaging, ADC = apparent diffusion coefficient.

**Figure 6 cancers-15-04536-f006:**
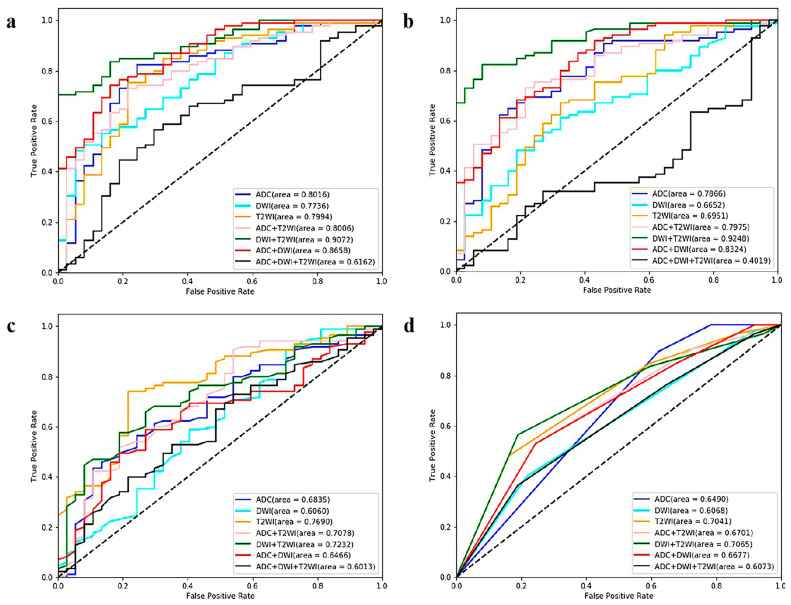
ROC curves of the twenty-eight ML models (based on four algorithms and seven sequences) for GGG prediction. (**a**) Logistic regression-, (**b**) support vector machine-, (**c**) random forest-, and (**d**) K-nearest neighbor-based model ROC curves. ROC = receiver operating characteristic, ML = machine learning, GGG = Gleason grade group, T2WI = T2-weighted imaging, DWI = diffusion-weighted imaging, ADC = apparent diffusion coefficient.

**Figure 7 cancers-15-04536-f007:**
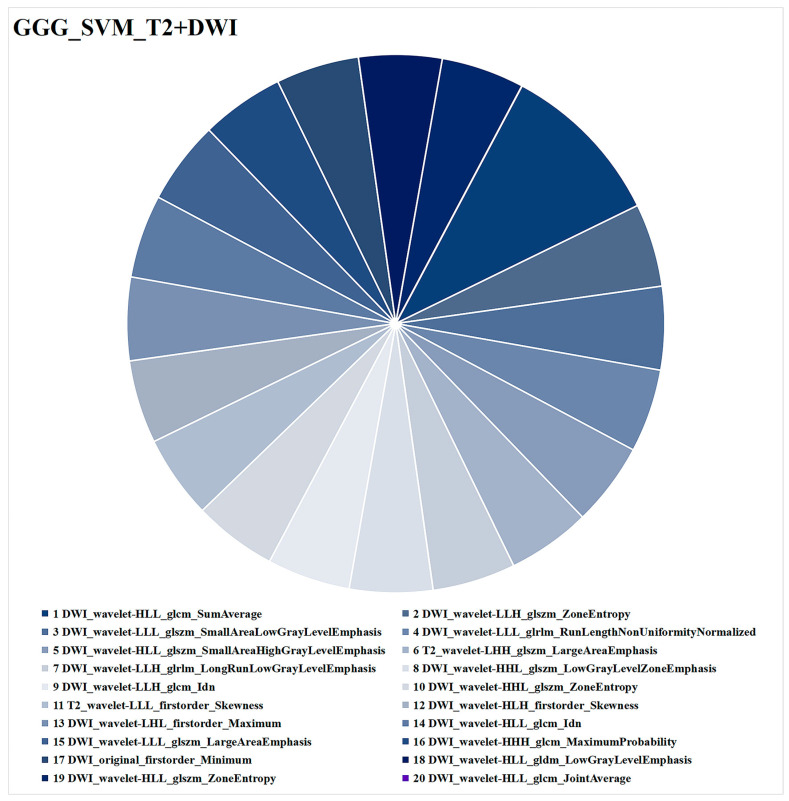
Optimal combined ML model (SVM_T2+DWI) for task 2 (GGG prediction). Different colors represent different radiomics features, and the area indicates the weighted contribution of each radiomics feature in the ML model. DWI_wavelet-HLL_glcm_SumAverage had the highest weight of all the radiomics features. ML = machine learning, SVM = support vector machine, T2WI = T2-weighted imaging, DWI = diffusion-weighted imaging, GGG = Gleason grade group.

**Figure 8 cancers-15-04536-f008:**
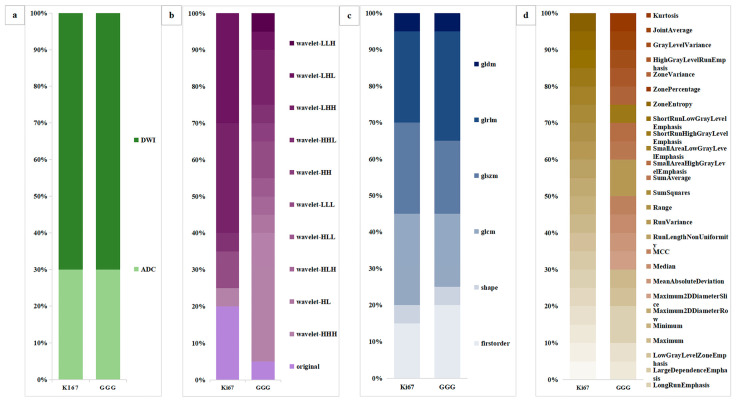
Comparative analysis of the radiomic features of LR_ADC+DWI. (**a**) The sources of the radiomics features. (**b**) Use of wavelet transformation for the radiomics feature. (**c**) Radiomics feature categories (first-order, shape, or texture). (**d**) Radiomics feature similarity between Ki67 expression and GGG prediction. LR = logistic regression, ADC = apparent diffusion coefficient, DWI = diffusion-weighted imaging, GGG = Gleason grade group.

**Table 1 cancers-15-04536-t001:** Ki67 prediction based on four machine learning algorithms and seven sequences *.

Images	LR			SVM			RF			KNN		
	AUC	Sens.	Spec.	AUC	Sens.	Spec.	AUC	Sens.	Spec.	AUC	Sens.	Spec.
T2	0.7383	0.6364	0.7463	0.7913	0.7091	0.7463	0.6946	0.5455	0.6567	0.6425	0.6000	0.6119
DWI	0.8578	0.7091	0.8209	0.8152	0.7455	0.7612	0.6408	0.5273	0.7910	0.6832	0.6182	0.7164
ADC	0.7715	0.7273	0.7612	0.7449	0.7273	0.6716	0.6689	0.5455	0.6866	0.5986	0.4364	0.6716
T2 + DWI	0.8285	0.7273	0.7761	0.7984	0.7455	0.7015	0.6634	0.5455	0.7015	0.6408	0.5455	0.6716
T2 + ADC	0.8882	0.7636	0.8657	0.8475	0.7091	0.7910	0.6965	0.5818	0.7164	0.6650	0.5818	0.7612
ADC + DWI	0.8516	0.7636	0.8209	0.7874	0.7091	0.7164	0.7115	0.5636	0.7015	0.6837	0.6000	0.7164
ADC + DWI + T2	0.8673	0.8182	0.8358	0.8336	0.8727	0.7463	0.6925	0.5636	0.7015	0.6794	0.5818	0.7015

* LR = logistic regression, SVM = support vector machine, RF = random forest, KNN = K-nearest neighbor, AUC = area under the curve, Sens. = sensitivity, Spec. = specificity, T2WI = T2-weighted imaging, DWI = diffusion-weighted imaging, ADC = apparent diffusion coefficient.

**Table 2 cancers-15-04536-t002:** GGG prediction based on four machine learning algorithms and seven sequences *.

Images	LR			SVM			RF			KNN		
	AUC	Sens.	Spec.	AUC	Sens.	Spec.	AUC	Sens.	Spec.	AUC	Sens.	Spec.
T2	0.7994	0.8706	0.5946	0.6951	0.7765	0.5946	0.7690	0.8824	0.4595	0.7041	0.8471	0.4054
DWI	0.7736	0.8824	0.4595	0.6652	0.7294	0.4324	0.6060	0.9059	0.2973	0.6068	0.7765	0.3243
ADC	0.8016	0.8706	0.5135	0.7866	0.8000	0.6216	0.6835	0.8824	0.6835	0.6490	0.8941	0.3784
T2 + DWI	0.9072	0.8706	0.6216	0.9248	0.8588	0.7838	0.7232	0.8000	0.3514	0.7065	0.8353	0.4054
T2 + ADC	0.8006	0.8471	0.4865	0.7975	0.7529	0.6757	0.7078	0.9059	0.4595	0.6701	0.9059	0.2703
ADC + DWI	0.8658	0.8824	0.5946	0.8324	0.7765	0.6757	0.6466	0.8235	0.2432	0.6677	0.8471	0.3243
ADC + DWI + T2	0.6162	0.7765	0.1892	0.4019	0.6353	0.3514	0.6013	0.8118	0.2973	0.6073	0.7647	0.3514

* GGG = Gleason grade group, LR = logistic regression, SVM = support vector machine, RF = random forest, KNN = K-nearest neighbor, AUC = area under the curve, Sens. = sensitivity, Spec. = specificity, T2WI = T2-weighted imaging, DWI = diffusion-weighted imaging, ADC = apparent diffusion coefficient.

**Table 3 cancers-15-04536-t003:** Prediction accuracy of the combined task *.

Images	LR	SVM	RF	KNN
T2	0.5080	0.4670	0.4340	0.4340
DWI	0.5820	0.4840	0.4920	0.4670
ADC	0.5660	0.5570	0.4590	0.4260
T2+DWI	0.5900	0.6230	0.4260	0.4430
T2+ADC	0.6070	0.5820	0.5250	0.4670
ADC+DWI	0.6230	0.5490	0.4020	0.4670
ADC+DWI+T2	0.5000	0.4340	0.3850	0.4020

* LR = logistic regression, SVM = support vector machine, RF = random forest, KNN = K-nearest neighbor, T2WI = T2-weighted imaging, DWI = diffusion-weighted imaging, ADC = apparent diffusion coefficient.

## Data Availability

Data are contained within the article.

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
