# Peer review of "MRI Radiomics-Based Machine Learning Models for Ki67 Expression and Gleason Grade Group Prediction in Prostate Cancer"

_cancers, 2023, doi:10.3390/cancers15184536_

Round 1
Reviewer 1 Report
Dear authors,
Thank you for raising such an interesting topic, regarding the potential role of MRI radiomics-based Machine Learning model in predicting Ki67 expression and Gleason grading group. However, there are some major issues that should be addressed by the authors.
- Title: According to task 2, the models were trained at differentiating patients with GGG 1-2 and 3-5. Though GGG is derived from GS, the title in its current form sounds misleading. It is pivotal to be clear and consistent regarding the objective of the study, does not use GGG and GS as interchangeable terms throughout the entire paper.
- Simple summary: ADT and GGG are not defined.
- Abstract: ADC is not a sequence.
Though positive and significant, the correlation between Ki67 and GS is a weak one, please specify (also in the results section and wherever is reported this result in the paper).
- Introduction: What do you mean with “By 2023”? According to the reference [1], the reported proportion is confirmed and it is not an expectation. Please revise accordingly.
The opposite of indolent pCA should be aggressive.
“Invasiveness” is related to the staging of the tumor; however, GS is a grading indicator. Please revise accordingly the definition of GS.
Please consider adding a reference for the following sentence “However, interobserver differences are a nonnegligible limitation 96 of the technique”.
- Methods: The models were trained using images derived from a bpMRI protocol. However, bpMRI is never mentioned in the paper and is not discussed (e.g., pros and cons vs mpMRI).
Fat saturation does not result in significant improvement over conventional T2-weighted images in endorectal MR imaging of the prostate (10.1016/0730-725x(94)92199-7). Considering the longer times required, why did you have included this sequence in your protocol?
How was the Ki67 threshold expression of 10% established?
Please specify the years of experience of the third reviewer.
The ROI positioning should be further detailed.
Please carefully use adjectives such as “robust”, particularly before selection and testing.
- Results: How many features were taken into account after feature elimination?
It is not clear where these results come from: the test set? The mean result from the 5-fold cross-validation? Please specify. If these results derive from the test set, please also add the results from the training set.
- Discussion: The discussion offer really interesting thoughts and insights about the results achieved by the authors. It could be interesting to discuss the added value offered by radiomics to bpMRI (10.3390/diagnostics12040799).
In this work, radiomics was used to predict prognostic markers (Ki67 and GGG). However, these markers are not without limitations, as pointed out by the authors themselves in the introduction. It would have been more appropriate to evaluate straightforwardly the potential prognostic role of features in terms of 5-os or DFS.
This is an important limitation of the work that should be emphasized.
The authors overlooked other significant limitations. In order to include all the limitation of the study, please carry out a self-assessment using the Radiomics Quality Score (RQS) (10.1038/nrclinonc.2017.141) and compare it to the mean achieved reported in literature specifically for prostate imaging (https://doi.org/10.1016/j.ejrad.2020.109095 ) and general one (https://doi.org/10.1007/s00330-022-09187-3).
Reviewer 2 Report
1. In chapter 2.4, the author mentioned ‘Radiomic data were extracted using PyCharm platform and the Scikit-learn software package’
PyCharm is a IDE (integrated development environment), you can write and compile code there, for example Python. But it is not a platform for radiomic feature extraction. Scikit-learn is a library to build and train machine learning model, neither for radiomic feature extraction. I doubt if the author actually did the experiments. The author need to specify which library/package did you use for radiomic feature extraction.
2. The author did not give enough information about the parameters for radiomic feature extraction, for example the resampling method and parameters, gray-level discretization method and bin size, what is the parameter of wavelet transformation used in image pre-processing and why did the author choose wavelet transformation instead of others. The values of radiomic features are highly influenced by these parameters. Without these information, it is difficult to reproduce the article.
3. The validation method is also unclear to me. The author mentioned the data set was divided into training set and validation set, then mentioned a 5-fold cross validation. Is the 20% data used as test set or validation set for model parameters optimization?
The author also used jargon that is unnecessary, the author also did not give a explanation to the terminology, such as reinforcement learning.
4. Is the reported final result based on validation or a test set blinded to the training set?
The writing is fluent and easy to follow, but try to avoid using unnecessary jargon, which you might also not understand yourself.
Round 2
Reviewer 1 Report
Dear Authors,
Thanks for having kindly accepted my suggestions, and congratulations for your very interesting work, that in my opinion is now suitable for publication in this eminent journal. Just a consideration: the acronym for Radiomics quality score is RQS and not QRS.
Author Response
Dear Authors,
Thanks for having kindly accepted my suggestions, and congratulations for your very interesting work, that in my opinion is now suitable for publication in this eminent journal. Just a consideration: the acronym for Radiomics quality score is RQS and not QRS.
Re: We thank the review again for the suggestions. We have revised the acronym for Radiomics quality score as RQS in the manuscript.